# The Role of Rehabilitation Nurses in Empowering Mastectomised Women for Self-Care: A Scoping Review

**DOI:** 10.3390/ijerph22060957

**Published:** 2025-06-18

**Authors:** Madalena Rodrigues, Inês Deus, Pedro Bengalinha, Raquel Duro, David Carpinteiro, Rogério Ferreira, Celso Silva, César Fonseca

**Affiliations:** 1São João de Deus School of Nursing, University of Évora, 7000-811 Évora, Portugal or m61663@alunos.uevora.pt (M.R.); or m61623@alunos.uevora.pt (I.D.); or m60993@alunos.uevora.pt (P.B.); or m60911@alunos.uevora.pt (R.D.); or m61720@alunos.uevora.pt (D.C.); cfonseca@uevora.pt (C.F.); 2Santa Maria Local Health Unit, Santa Maria Hospital, 1649-028 Lisboa, Portugal; 3Lisbon Centre for Research, Innovation and Development in Nursing (CIDNUR), 1600-096 Lisboa, Portugal; 4Alentejo Central Local Health Unit, Évora Espírito Santo Hospital, 7000-811 Évora, Portugal; 5Algarve Local Health Unit, Faro Hospital, 8000-386 Faro, Portugal; 6Litoral Alentejano Local Health Unit, Litoral Alentejano Hospital, 7540-230 Santiago do Cacém, Portugal; 7School of Health, Polytechnic Institute of Beja, 7800-295 Beja, Portugal; ferrinho.ferreira@ipbeja.pt; 8Comprehensive Health Research Centre—CHRC-UÉ, University of Évora, 7000-811 Évora, Portugal

**Keywords:** rehabilitation, nurse, intervention, mastectomy, quality of life

## Abstract

Background: Breast cancer is one of the most prevalent neoplasms among women, often requiring mastectomy, a procedure with a significant impact on functionality, self-esteem, and quality of life. Objective: This study aimed to map the main interventions performed by the Rehabilitation Nursing Specialist in the follow-up care of mastectomised women. Methods: A review was conducted according to the Joanna Briggs Institute methodology. The search included 11 articles published between 2019 and 2024 in Portuguese, English, and Spanish, available on the EBSCO platform (MEDLINE with Full TEXT, CINAHL). The descriptors used were (Mastectomy OR Breast Removal) AND (Rehabilitation Nursing OR Nursing Intervention). Results: The Rehabilitation Nursing Specialist interventions focused on education regarding upper limb mobilisation, medication administration, lymphedema prevention, strategies for performing Daily Life Activities with less effort and pain, implementation of rehabilitation plans with physical exercises, and emotional support in accepting body image changes. Hospital discharge planning and caregiver education also emerged as key elements to ensure continuity of care. Most studies (six) identify performing exercises to strengthen muscles and prevent lymphoedema as a very important intervention for nurses, followed by education on care for upper limb mobilisation and lymphoedema prevention (five) and emotional and social support (four), among other interventions. Conclusions: The Rehabilitation Nursing Specialist interventions are fundamental for promoting functionality, emotional well-being, and quality of life in mastectomised women, reinforcing the importance of a multidisciplinary, person-centred approach supported by scientific evidence.

## 1. Introduction

Breast cancer is the most common malignant neoplasm among women worldwide and represents one of the leading causes of cancer death in the female population. According to the World Health Organisation (WHO) [1], over 2.3 million new cases of breast cancer were diagnosed in 2020, resulting in approximately 685,000 deaths globally. This disease represents a significant public health problem with substantial physical, emotional, and social implications.

In Portugal, breast cancer accounts for approximately 30% of all cancer cases in women [2]. Mastectomy involves the removal of the breast, usually associated with axillary dissection [2], and is the most common method used in breast cancer treatment [3]. The breast is a gland composed of various structures, including lymphatic vessels. The surgery may be performed to remove the tumour, screen for axillary lymph node invasion, or alleviate symptoms in advanced-stage disease [4].

As a consequence of this surgical procedure, women develop alterations in self-image and self-esteem, as well as fear regarding these bodily changes and the impact they may have on their social and professional lives [5]. Rehabilitation nursing can significantly improve health-related quality of life for patients with chronic diseases by addressing complex care needs and improving patient coping strategies, ultimately leading to better management of chronic conditions and improved patient outcomes [6,7].

Rehabilitation nurses play a crucial role in healthcare organisations, requiring a diverse set of skills to effectively support patients. The changing healthcare landscape emphasises not only technical proficiency but also cultural competence, effective communication, and interprofessional collaboration. Culturally competent Rehabilitation Nurses should possess skills, knowledge, and attitudes that enhance understanding of patients‘ values, awareness of personal biases, and the ability to tailor care to patients’ expectations and preferences, ensuring effective and respectful healthcare delivery [8].

Rehabilitation nurses must have skills in effective communication, interprofessional collaboration, the use of supportive technology, providing patient- and family-centred care and promoting self-management, which includes evidence-based practice, collaboration in interprofessional teams, and proficiency in assessing and improving the quality of care, all aimed at improving patients’ quality of life [9,10]. These competencies are essential for promoting safe and effective transitions of care in various healthcare settings and improving patient outcomes [9,11].

Functional re-education is a standard of quality care defined by the Order of Portuguese Nurses (https://www.ordemenfermeiros.pt/, accessed on: 1 March 2025), as well as one of the quality standards for specialised care in rehabilitation nursing. Therefore, the Rehabilitation Nurse, together with the care recipient, should develop functional re-education processes aimed at self-care, quality of life, reintegration, and societal participation [12,13]. Upper limb rehabilitation nursing programmes positively impact quality of life in post-mastectomy breast cancer patients, improving functioning and symptoms [14,15], as well as improving quality of life [16].

There are not many studies that more clearly identify the role of Rehabilitation Nurses in training mastectomised women in self-care. Our study aims to address this gap.

The aim of this study was to map the main interventions carried out by the Rehabilitation Nurse Specialist in the follow-up of mastectomised women.

This systematic literature review is registered on the PROSPERO platform under registration number CRD420250652280.

## 2. Methodology

### 2.1. Study Design

The proposed review was conducted according to the updated methodology of the Joanna Briggs Institute (JBI) [17] and consisted of the following steps:

#### 2.1.1. Research

Research in MEDLINE with Full TEXT and CINAHL databases.

#### 2.1.2. Research Question

The question guiding this review was, “What are the interventions of the Rehabilitation Nurse in empowering mastectomised women for self-care?”.

#### 2.1.3. Strategy PCC

The PCC strategy from the Joanna Briggs Institute [17] was used, where the following applies:Population (P): Mastectomised women;Concept (C): Interventions of the Rehabilitation Nurse;Context (C): Empowerment for self-care.

#### 2.1.4. Inclusion Criteria

Only studies available in full text, published in Portuguese, English, or Spanish, between 1 January 2019 and 31 December 2024, were included to obtain the recent and updated literature. This time limit was considered appropriate given the objectives of the review and the question that guided it [18]. This is because, in recent years, there has been a greater increase in the role of the Rehabilitation Nurse for a few reasons, including medical and technological advances in the surgical approach, a greater focus on quality of life and patient autonomy, and patient education and self-management where the role of the Rehabilitation Nurse has been emphasised. We also wanted to avoid reporting older or outdated practices compared to current ones.

Studies that did not fulfil the inclusion criteria were excluded. Two independent reviewers assessed the eligibility of the studies. If there was any disagreement about the inclusion and/or exclusion of a study, a third reviewer was consulted.

#### 2.1.5. Study Types

All types of articles were considered: studies with quantitative, qualitative, and mixed methodologies, including primary studies and reviews.

#### 2.1.6. Search Strategy

We carried out the search using the terms DeCS/MESH used as descriptors: (“Mastectomy” OR “Breast Removal”) AND (“Rehabilitation Nursing” OR “Nursing Intervention”). After the search, all identified articles were imported into the Mendeley (v2.134.0)^®^ reference manager, and duplicates were removed.

#### 2.1.7. Study Selection Process

According to Levac, Colquhoun, and O’Brien (2010) [19], two independent reviewers assessed the titles and abstracts based on the inclusion criteria. Selected articles were subsequently analysed in full text. In case of doubt regarding the inclusion of any article, a third reviewer was consulted.

#### 2.1.8. Data Analysis and Extraction

The results of the different stages of the selection process were organised according to the PRISMA Flow Diagram model [20]. Data extraction from each article was performed using a grid that included the following parameters: author, year of publication, country of origin, study type, study location, objectives, methodology, population/number of participants, definition of Rehabilitation Nurse, definition of mastectomy, main interventions, main results, and conclusions.

#### 2.1.9. Data Synthesis and Evaluating the Quality of Studies

The review presented a set of interventions by the Rehabilitation Nursing Specialist obtained through the analysis and interpretation of the included articles.

A descriptive synthesis was prepared, highlighting the most relevant interventions in the process of empowering mastectomised women for self-care, aiming to improve their quality of life. A descriptive analysis of each study was carried out using a form designed for this purpose. This form underwent a pilot test, where it was found to be a suitable tool and enabled the information to be extracted in accordance with the research question. Data extraction was carried out independently by the reviewers responsible for selecting the studies, and in the event of a disagreement, another reviewer was consulted. The data extracted contains specific details on authors, years, titles, objectives, study design, methods, results, and conclusion.

The quality assessment of the studies to be included in this review followed the critical appraisal tools of the JBI, which help assess the reliability, relevance, and results of published articles, and the checklist was completed according to the type of study [21]. Two independent reviewers assessed the studies based on the checklist criteria. In case of doubt, a third reviewer was consulted.

We chose to draw up a synthesis of all the nursing interventions that the selected studies described as appropriate, i.e., those that promote the empowerment of mastectomised women for self-care. The reader has access to all the interventions reported in the selected studies and can choose, if they wish, to implement the ones they consider most appropriate in the context of their clinical practice.

Synthesising the data in this review involved analysing and interpreting the 11 articles selected to identify the most relevant interventions carried out by the Rehabilitation Nurse Specialist in the process of empowering mastectomised women to self-care. The main aim of the synthesis was to highlight the interventions that contribute to improving the quality of life of these women.

## 3. Results

The search was carried out in the MEDLINE with Full TEXT and CINAHL databases, identifying 523 articles. From a total of 523 studies, 63 duplicate studies were removed, leaving 460. Of the 460 studies, 263 were excluded by reading the title, and 184 studies were excluded by reading the abstract. Of the 13 studies selected, 2 were also excluded because they did not fit the theme. The final number of studies selected was 11 (7 in Portuguese and 4 in English) (Figure 1. PRISMA flow diagram of the study selection process).

To facilitate understanding and application of the interventions, the data were organised and presented in Table 1 (Description of included studies). This table provides an overview of each study, including references, objectives, specific interventions, and level of evidence.

In terms of quality assessment, all 11 studies had a low risk of bias/high quality, i.e., the studies met most or all the methodological criteria in the respective JBI checklist depending on the type of study, and the decision was made to include them in the review [32].

The interventions were categorised based on recurring themes in the articles, such as education, promotion of functionality, prevention of complications, and psychosocial support. The synthesis sought to present all the interventions that the selected studies described as suitable for promoting self-care training for mastectomised women.

After analysing the 11 articles presented in Table 1, we were able to define common interventions such as educating the woman undergoing mastectomy, empowerment, and promoting self-care.

Promoting the woman’s functionality and autonomy is achieved by providing strategies to perform ADL with less effort and pain and adapting tasks to ensure the patient’s independence in daily life.

Recommending practices that respect the patient’s autonomy and values: We also highlight the implementation of physical exercise plans for lymphedema prevention, post-surgery, and when at home, planning the discharge with the woman undergoing the surgical procedure and her family/caregivers.

Emotional support, the implementation of strategies to promote self-image and self-confidence, and the creation of discussion groups are interventions present in the various analysed articles.

All interventions, as shown in Table 2, must always consider the individual’s uniqueness, use a multidisciplinary approach, and be applicable in all phases of the pre-operative, peri-operative, and post-operative processes.

## 4. Discussion

Mastectomy is considered the main treatment for breast cancer and brings physical and psychological changes for women, requiring a coordinated and holistic intervention by health professionals [3].

Empowerment and the promotion of self-care appear as central themes in several studies, aiming to maintain and promote functionality by finding strategies for performing daily life activities with less pain and effort [23,24,29]. Our results are in line with previous studies regarding very important dimensions in the provision of post-mastectomy care [33,34,35]. In fact, promoting self-care has numerous benefits for physical, mental, and emotional health, improving quality of life, productivity, relationships, and self-esteem. Self-care helps to reduce stress and anxiety, as well as increase resilience and well-being, which are very important for mastectomised women.

To empower the woman undergoing mastectomy, their participation in the process of preventing complications, recovery, and rehabilitation should begin immediately after surgery, addressing strategies aimed at maintaining the woman’s functionality [28].

Following surgery, there is a reduction in the range of motion of the shoulder joint, making the adaptation of tasks to ensure the woman’s independence in performing daily life activities significantly important [15,23,24,25]. It is, therefore, crucial to perform an assessment of the shoulder joint’s range of motion and the muscle strength of the affected upper limb post-mastectomy using goniometry and dynamometry, respectively [31]. In the same vein, previous studies have emphasised that assessing the range of motion of the shoulder joint, with the aim of restoring functionality as much as possible, is important for promoting self-care and autonomy [36,37,38]. We consider this aspect to be very relevant, as improving shoulder range of motion and muscle strength has a positive impact on women’s functionality and on regaining their autonomy.

Previous studies have emphasised the role of rehabilitation nursing in functional rehabilitation programmes for the shoulder joint, which not only restores function to patients but also improves their quality of life [39,40]. Rehabilitation nursing significantly improves the quality of life of post-mastectomy breast cancer patients by addressing the physical, psychological, and social consequences, promoting self-confidence, and facilitating social connections, ultimately helping with the transition to normal daily life and improving general well-being [41].

Thus, a personalised exercise plan should be developed and implemented, considering needs and preventing potential complications [22,25,30,31] through joint mobilisation, stretching, and exercises aimed at muscle strengthening [30]. Rehabilitation nursing significantly improves the quality of life of post-mastectomy breast cancer patients. One study reports that most patients reported positive influences on their daily functioning and general well-being, with 92% recognising the beneficial impact of rehabilitation on their quality of life [42].

Empowering women to perform these exercises at home should be considered an extremely important intervention by the Rehabilitation Nursing Specialist, requiring teaching and training the woman to correctly follow the developed plan [23,29]. These exercises, besides aiming to improve the range of motion and, consequently, the functionality of the affected upper limb, also aim to reduce and prevent lymphedema [29].

Lymphedema, thus, emerges as the main complication associated with mastectomy, causing changes in body image, discomfort, and reducing the independence of the mastectomised woman. It is, therefore, necessary to address strategies that minimise these effects. Previous studies have reported the advantages of training women in exercises that allow them to increase the range of motion of the shoulder joint and, thus, recover as much function as possible [43,44].

Therefore, the use of compression sleeves, performing lymphatic drainage, as well as performing and teaching exercises aimed at reducing oedema can be approaches used to minimise these consequences [15,23,31]. Detailed explanations should be provided to women about specific practices and guidelines such as mobilisation, the use of assistive devices, and methods for lymphedema control [15,23,24,25,31]. Lymphoedema greatly limits a woman’s functionality and autonomy and, consequently, interferes with self-care. Therefore, interventions to reduce or eliminate lymphoedema are very important when caring for these women.

Rehabilitation nursing programmes have led to significant improvements in physical health indicators, such as increased range of movement in the upper limbs and a reduction in lymphoedema, as well as a reduction in fatigue and an increase in functionality [16,45].

In addition, given the possible physical alterations previously mentioned resulting from mastectomy, intervening in the psychological and emotional recovery and well-being of the mastectomised woman should be considered a priority by the Rehabilitation Nursing Specialist [22], which was already recommended in previous studies [46,47].

The fear of losing her role as a woman and, if applicable, as a mother, as well as the inherent fear of dying, are frequent thoughts among these women, increasing the prevalence of anxiety in this population [24]. Therefore, reducing anxiety using relaxation methods can be equally important in maintaining the psychological well-being of these women [22].

In this sense, active listening and welcoming, encouraging self-reflection, defining strategies to aid self-image acceptance, and referring to support groups where topics such as self-acceptance, sexuality, and even the breast cancer diagnosis itself are discussed are interventions performed by the RNS aimed at improving psychological well-being and promoting body acceptance, reducing complications that could arise from them [23,24,25,28,29]. In fact, considering the changes in self-image that mastectomy causes, interventions aimed at improving psychological well-being, in addition to physical well-being, can promote the mental health of these women and prevent depressive symptoms associated with self-image and possible physical limitations.

We infer, then, that the empowerment and education of mastectomised women assume great importance in the rehabilitation process, providing adaptive strategies that promote the maintenance of functionality and self-care, preventing physical complications, and promoting a safe environment so that women can maintain the highest possible level of autonomy and independence in ADL. Figure 2 summarises the interventions carried out by rehabilitation nursing.

We identified some limitations. Although the included studies had good methodological quality, they did not clarify some important aspects, such as the frequency of nurses’ interventions and how long they were implemented.

The implications for clinical practice of the results and subsequent discussion of this review are as follows: Multidisciplinary and person-centred approach: It is essential that Rehabilitation Nurses work as part of a multidisciplinary team, ensuring that the care provided is individualised and adapted to the needs of each mastectomised woman. The promotion of self-care and functionality: Interventions should focus on empowering women to self-care through education on upper limb mobilisation, medication administration, and the prevention of lymphoedema. The aim is for women to perform Activities of Daily Living (ADL) with less effort and pain, promoting their independence. The implementation of physical rehabilitation plans: It is crucial to develop and implement personalised physical exercise plans for muscle strengthening and lymphoedema prevention, both post-operatively and at home. Assessment of shoulder joint range of motion and muscle strength is essential for regaining functionality. Psychosocial and emotional support: Emotional support is a key intervention to help women cope with changes in body image, acceptance of self-esteem, and a reduction in anxiety. Active listening, welcoming, and encouraging self-reflection and referral to support groups are important practices. Hospital discharge planning: The Rehabilitation Nurse should actively participate in hospital discharge planning together with the woman and her family/carers, ensuring continuity of care and education about post-operative care and warning signs. Education for family members/carers: Educating family members and carers is a key element in ensuring continuity of care and support for women in their recovery process. Use of scientific evidence: Interventions must be based on scientific evidence in accordance with the guidelines and quality standards defined by the Portuguese Order of Nurses for Rehabilitation Nursing. Intervention at all stages of the process: Interventions should be applied at all stages of the pre-operative, peri-operative, and post-operative processes, with a view to early rehabilitation and effective management of suffering.

## 5. Study Limitations

This review only included studies in English, Portuguese, and Spanish, so linguistic bias may have affected the comprehensiveness of our review. Likewise, only studies from 2019 to 2024 were included, so the results may be limited by the time frame; in other words, this time frame may have limited the scope of the evidence.

## 6. Conclusions

The Rehabilitation Nurse Specialist plays a decisive role in empowering mastectomised women, promoting self-care, and contributing to a full recovery, thus increasing quality of life.

It is concluded that the nurse is a fundamental piece in the recovery and rehabilitation of mastectomised women. The RNS acts as a facilitator for improving quality of life, promoting not only physical recovery but also self-acceptance and emotional support for women. These professionals play an essential role in organising and managing care, highlighting their importance in the multidisciplinary team.

Furthermore, evidence indicates that strategies such as the use of devices and multidisciplinary approaches expand the possibilities for social reintegration and the recovery of self-esteem. Therefore, it is crucial that rehabilitation care is based on evidence-based practices, as suggested by the College of the Specialty of Rehabilitation Nursing of the Order of Portuguese Nurses (https://www.ordemenfermeiros.pt/a-ordem/col%C3%A9gios/mcee-de-reabilita%C3%A7%C3%A3o/, accessed on 1 March 2025). The RNS, as a facilitating agent, plays a vital role in organising and implementing patient-centred interventions, contributing to strengthening their self-confidence and autonomy.

The texts converge in emphasising that the Rehabilitation Nurse possesses crucial competencies for the comprehensive care of mastectomised women. This professional is responsible for developing strategies that promote early rehabilitation, the effective management of suffering, a reduction in complications, and the recovery of quality of life. Moreover, emotional support, family integration, and empowerment for self-care reinforce the positive impact of nursing interventions, demonstrating the relevance of individualised, evidence-based programs.

## Figures and Tables

**Figure 1 ijerph-22-00957-f001:**
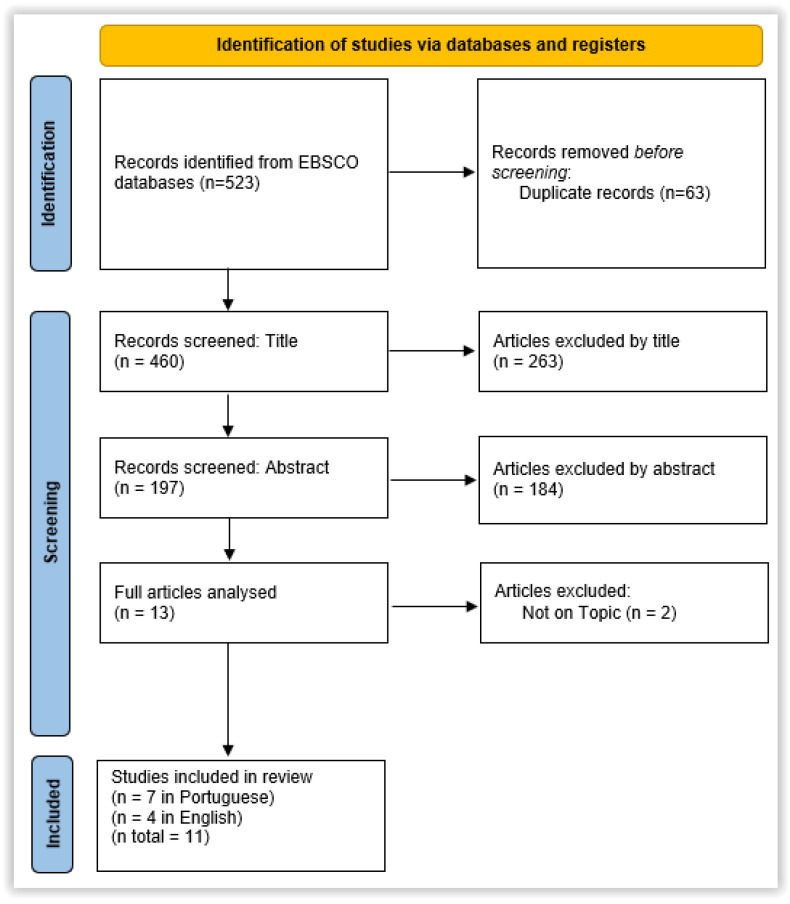
PRISMA flow diagram of the study selection process.

**Figure 2 ijerph-22-00957-f002:**
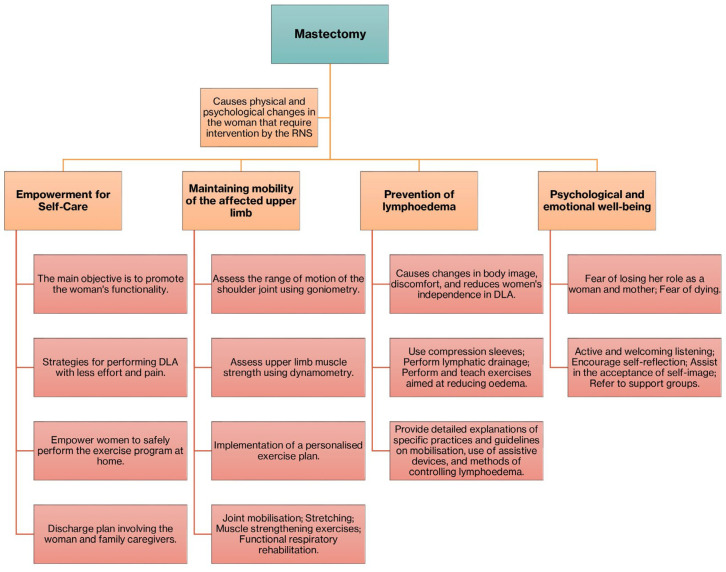
Summary diagram of interventions carried out by rehabilitation nursing.

**Table 1 ijerph-22-00957-t001:** Description of the included studies.

References	Objectives	Interventions	Level of Evidence
[22]	Recognise the importance of the nurse in the post-operative period to assist women in rehabilitation after mastectomy with axillary lymph node dissection.	- Education for mastectomised women on shoulder mobilisation, medication administration, weightlifting, and anxiety control. - Promotion of self-care, encouraging physical exercise and skin care. - Psychological support through listening and welcoming to help women cope with body image changes, define strategies for self-image acceptance, involve family, develop individualised care plans, and guide towards community support groups.	2.b Systematic review of quasi-experimental designs
[23]	Describe the experiences of mastectomised women in the home setting. Identify the importance attributed by women to the intervention of the Rehabilitation Nurse.	- Empower, educate, and teach about exercises to improve the range of motion and reduce oedema, the use of devices like compression sleeves, and skin care. - Promotion of functionality using strategies to perform ADL with less pain and effort, and task adaptation to facilitate independence. - Emotional support promoting acceptance and self-confidence.	3 Single qualitative study
[24]	Identify nursing actions that restore self-esteem in patients undergoing mastectomy.	- Prevent complications secondary to surgery and promote independence in functional capacity recovery. - Perform physical exercises and lymphatic drainage exercises. - Emotional support for prevention of psychological and body image-related complications. - Encourage self-reflection. - Support return-to-life activities, social, leisure, work, and family. - Implement respiratory rehabilitation programs.	3.b Systematic review of comparable cohorts
[25]	Recognise how mastectomised women describe and understand the nursing care received in a rehabilitation centre.	- Education on lymphedema prevention. - Perform physical exercises aimed at preventing and controlling complications with the arm and shoulder after surgery. - Promote social and leisure activities (such as discussion groups).	3 Single qualitative study
[26]	Map the use of exergames in the rehabilitation of individuals undergoing breast surgery, identifying health gains from implementing this technology.	- Functional exercises associated with exergames.	1.b Systematic review of RCT
[15]	What is the relationship between implementing a rehabilitation program and the self-care performance capacity in women undergoing breast surgery with lymph node dissection?	- Education on lymphedema risk-reduction measures, upper limb and cervical spine mobilisation exercises, and scar massage. - Application of a daily 45 min program for 3 months.	2.c Quasi-experimental prospectively controlled study
[27]	Determine the value of using Neuman’s Systems Model in providing care to women undergoing mastectomy.	- Assessment of stressors and their source pre-operatively, peri-operatively, and post-operatively. - Education on the surgical plan and possible complications pre-operatively. - Education on post-operative care.	1.d Pseudo-RCTs
[28]	Analyse scientific productions from the last 10 years addressing actions and guidelines on the discharge plan for women undergoing mastectomy.	- Promotion of self-care, considering the participation of the mastectomised woman herself in the process of preventing complications, recovery, and rehabilitation after surgery. - Education for the woman and family/caregivers about post-operative care and warning signs. - Formulate discharge plans with the woman and family/caregivers. - Emotional support.	3.b Systematic review of comparable cohorts
[29]	Describe the experience lived in an interdisciplinary follow-up centre for mastectomised women at a public university in São Paulo during the beginning of the COVID-19 pandemic.	- Physical exercise for lymphedema reduction and prevention. - Group therapy. - Teaching exercises to be performed at home. - Clarify doubts the woman may have about the process.	3.d Case—controlled study
[30]	Identify functionality indicators and types of interventions used for functional assessment and rehabilitation of the upper limb in post-mastectomy women.	- Perform joint mobilisation, stretching, and muscle strengthening.	1.b Systematic review of RCT
[31]	Identify and describe conservative interventions and clinical outcome assessment tools used in the peri-operative physical rehabilitation of women with breast cancer awaiting or having undergone mastectomy.	- Patient education. - Performance of physical exercise. - Performance of lymphatic drainage. - Assessment of shoulder joint range of motion. - Assessment of muscle strength.	2.b Systematic review of quasi-experimental designs

**Table 2 ijerph-22-00957-t002:** Identified interventions.

Interventions:	Articles:
Education on care for upper limb mobilisation and lymphedema prevention.	[15,22,23,25,31]
Promotion of self-care.	[22,28]
Empowerment for return to daily life activities.	[23,24]
Education for family/caregivers.	[22,28]
Education, empowerment, and promotion of physical exercise.	[23,29]
Emotional and social support.	[22,24,27,28]
Performance of exercises for muscle strengthening and lymphedema prevention.	[25,26,28,29,30,31]
Functional respiratory rehabilitation.	[24]
Assessment of muscle strength and joint range of motion.	[24]
Discharge plan with the woman and family/caregivers.	[28]

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
