# Peer review of "The Role of Rehabilitation Nurses in Empowering Mastectomised Women for Self-Care: A Scoping Review"

_ijerph, 2025, doi:10.3390/ijerph22060957_

Round 1

Reviewer 1 Report

Comments and Suggestions for Authors

This manuscript explores the interventions implemented by Rehabilitation Nursing Specialists to empower mastectomized women for self-care. The topic is highly relevant and timely, particularly in the context of comprehensive breast cancer management. While the paper has many strengths, some aspects require refinement. Below I will provide a section-by-section report, highlighting strengths and areas for improvement.

Abstract: The abstract is concise and provides a valuable summary of the study, but it could benefit from the report of quantitative results.

Introduction: This section provides a good rationale for the study based on the global and national burden of breast cancer, and it effectively frames the importance of rehabilitation nurses in post-mastectomy care. Nonetheless, the background could benefit from a more critical analysis of current gaps in the literature to better position the added value of this review.

Methods: The strategy used is methodologically sound, and inclusion and exclusion criteria are well-defined. However, I noticed some gaps that the authors should address: a) the search strings are not reported; b) quality assessment is not mentioned; c) the PRISMA diagram is referenced but not fully detailed within the manuscript; and d) the rationale for limiting the literature to studies from 2019–2024 should be more explicitly justified.

Results: Table 1 and 2 are well-organized and informative. From the results, we acknowledge how the nature of rehabilitation nurses is multidisciplinary, given that the included studies span a wide range of interventions. However, the qualitative reporting of results could benefit from thematic clustering of interventions and an indication of how frequently each intervention type appears across studies. Moreover, the evidence classification system used (e.g., what 1.a, 2.b mean) should be clarified.

Discussion and Conclusions: The Discussion appropriately reflects on the clinical relevance of findings, but there is a tendency to restate the results rather them to interpret them. The authors could consider implementing this section by addressing limitations in the current evidence, or gaps, and to discuss the quality of the included studies. The Conclusions effectively emphasizes the centrality of the rehabilitation nurses in post-mastectomy care.

Author Response

Dear Reviewer 1,

Thank you very much for your very useful comments. We will respond to them and we think that the manuscript has been improved by your suggestions. We will respond point by point in order to improve the manuscript. Please let us know if any further changes are needed.

Comment 1: Abstract: The abstract is concise and provides a valuable summary of the study, but it could benefit from the report of quantitative results.

Response 1: Thank you very much for your comment and suggestion.
We've added some more data to the summary. Lines 33-36.

Comment 2: Introduction: This section provides a good rationale for the study based on the global and national burden of breast cancer, and it effectively frames the importance of rehabilitation nurses in post-mastectomy care. Nonetheless, the background could benefit from a more critical analysis of current gaps in the literature to better position the added value of this review.

Response 2: Thank you very much for your comment. We agree with a more critical analysis. We have added this concern. Lines 85-86.

Comment 3: Methods: The strategy used is methodologically sound, and inclusion and exclusion criteria are well-defined. However, I noticed some gaps that the authors should address: a) the search strings are not reported; b) quality assessment is not mentioned; c) the PRISMA diagram is referenced but not fully detailed within the manuscript; and d) the rationale for limiting the literature to studies from 2019–2024 should be more explicitly justified.

Response 3: Thank you very much for your warnings. In response to your request, we have added lines 107-109; 133; 139-143; 146-150; and reformulated the prism diagram to make it clearer for the reader. 

Comment 4: Results: Table 1 and 2 are well-organized and informative. From the results, we acknowledge how the nature of rehabilitation nurses is multidisciplinary, given that the included studies span a wide range of interventions. However, the qualitative reporting of results could benefit from thematic clustering of interventions and an indication of how frequently each intervention type appears across studies. Moreover, the evidence classification system used (e.g., what 1.a, 2.b mean) should be clarified.

Response 4: Thank you very much for your comment. Only one study indicates a frequency (application of a 45-minute daily programme for 3 months). The other studies don't specify a frequency, which leads us to assume that it could be a continuous frequency or depending on the interaction with the patient. Therefore, if you agree, we would keep it as it is in the manuscript. We have clarified the evidence classification system in table 1. In any case, we've added lines 154-163; 172-178.

Comment 5: Discussion and Conclusions: The Discussion appropriately reflects on the clinical relevance of findings, but there is a tendency to restate the results rather them to interpret them. The authors could consider implementing this section by addressing limitations in the current evidence, or gaps, and to discuss the quality of the included studies. The Conclusions effectively emphasizes the centrality of the rehabilitation nurses in post-mastectomy care.

Response 5: Thank you very much for your comment. We've added lines 204-207; 218-220; 251-253; 271-274; 283-285.

Thank you very much for your work.
Best regards.

Reviewer 2 Report

Comments and Suggestions for Authors

Congratulations on the valuable work. Your study makes a significant contribution to such a sensitive area concerning the care of women who have undergone mastectomy. However, some minor revisions could be made, mainly in the discussion section, to make its contribution to the literature even more impactful.

My observation is that there is a lack of deeper analysis of the studies, which are mostly presented descriptively. Clearly, this section should summarize all the relevant studies on the topic but with a critical perspective and an analysis of gaps, controversies, and unresolved questions in order to better frame both the contribution and the challenges of the present study.

Author Response

Dear Reviewer 2,

Thank you very much for your very useful comments. We will respond to them and we think that the manuscript has been improved by your suggestions. We will respond point by point in order to improve the manuscript. Please let us know if any further changes are needed.

Comment 1: 

Congratulations on the valuable work. Your study makes a significant contribution to such a sensitive area concerning the care of women who have undergone mastectomy. However, some minor revisions could be made, mainly in the discussion section, to make its contribution to the literature even more impactful.

My observation is that there is a lack of deeper analysis of the studies, which are mostly presented descriptively. Clearly, this section should summarize all the relevant studies on the topic but with a critical perspective and an analysis of gaps, controversies, and unresolved questions in order to better frame both the contribution and the challenges of the present study.

Response 1: Thank you very much for your very useful comments. We will respond to them and we think that the manuscript has been improved by your suggestions. We have added clarifications that we think are in line with your comments. Please let us know if anything else is needed. We've added lines in the discussion section: We've added lines 204-207; 218-220; 251-253; 271-274; 283-285. 

Thank you very much for your work.
Best regards.

Reviewer 3 Report

Comments and Suggestions for Authors

Dear Authors,

Thank you for the opportunity to review your manuscript titled "Rehabilitation Nursing Interventions for Women Post-Mastectomy: A Systematic Review." Your work addressing the role of nursing interventions in post-mastectomy rehabilitation is both timely and relevant, contributing valuable insights to the field.

However, after a thorough examination of the manuscript, I have several suggestions that I believe would enhance its clarity, methodological rigour, and overall quality:

  1. Addressing Limitations
  • Please explicitly discuss the potential for linguistic bias due to the inclusion criteria limited to studies published in English, Portuguese, and Spanish. Clarifying how this may have impacted the comprehensiveness of your review would strengthen the manuscript, particularly if there were any cultural influences on the therapies reported.
  • Moreover, it will be extremely helpful for readers to know how many of these articles were published in these three languages. Please include the information on the PRISMA flow diagram.
  • Additionally, acknowledge the implications of restricting the literature search to studies published between 2019 and 2024 and how this temporal scope might limit the scope of evidence.
  1. Methodological Clarity and Rigor
  • Provide more detailed information on the quality assessment of the included studies. Including a methodological appraisal (e.g., using standardised tools such as Cochrane risk-of-bias tools) would allow readers to better understand the strength of the evidence underpinning your conclusions.
  • Clarify how you managed the heterogeneity arising from different study designs (quantitative, qualitative, mixed methods) in your synthesis process. Similar to the earlier suggestion, please provide information on how many articles were identified based on the study designs. Discuss any limitations this heterogeneity may impose on the interpretation of your findings.
  • If applicable, consider elaborating on the criteria used for study inclusion/exclusion and how disagreements among reviewers were resolved to ensure transparency. I noticed that little was mentioned regarding the exclusion criteria.
  1. Presentation and Completeness
  • Expand the data synthesis section to detail how interventions were categorised and synthesised across studies. Providing a clearer narrative on the main themes and types of interventions would improve comprehensiveness.
  1. Language and Style
  • Minor language editing is recommended to improve clarity and style throughout the manuscript. This will enhance readability and ensure your findings are conveyed effectively.
  • DLA: there is a lack of explanation on the term DLA, of which I presumed it stands for Daily living activities. If so, the more commonly used term should be Activities of Daily Living (ADL).
  • It is rather unconventional for the Limitation to come after the Conclusion, since conclusion implies the end of manuscript.

In summary, I appreciate the importance of your study and believe that addressing these points will significantly strengthen your manuscript, making it more robust and valuable to clinicians and researchers alike. I look forward to your revised submission.

Thank you for your consideration.

Author Response

Dear Reviewer 3,

Thank you very much for your very useful comments. We will respond to them and we think that the manuscript has been improved by your suggestions. We will respond point by point in order to improve the manuscript. Please let us know if any further changes are needed.

Comment 1: 

  1. Addressing Limitations
  • Please explicitly discuss the potential for linguistic bias due to the inclusion criteria limited to studies published in English, Portuguese, and Spanish. Clarifying how this may have impacted the comprehensiveness of your review would strengthen the manuscript, particularly if there were any cultural influences on the therapies reported.
  • Moreover, it will be extremely helpful for readers to know how many of these articles were published in these three languages. Please include the information on the PRISMA flow diagram.
  • Additionally, acknowledge the implications of restricting the literature search to studies published between 2019 and 2024 and how this temporal scope might limit the scope of evidence.

Response 1: Thank you very much for your comment. We've added lines 287-290. We've updated the PRISMA flow chart. We reviewed lines 287-290.

Comment 2: 

  1. Methodological Clarity and Rigor
  • Provide more detailed information on the quality assessment of the included studies. Including a methodological appraisal (e.g., using standardised tools such as Cochrane risk-of-bias tools) would allow readers to better understand the strength of the evidence underpinning your conclusions.
  • Clarify how you managed the heterogeneity arising from different study designs (quantitative, qualitative, mixed methods) in your synthesis process. Similar to the earlier suggestion, please provide information on how many articles were identified based on the study designs. Discuss any limitations this heterogeneity may impose on the interpretation of your findings.
  • If applicable, consider elaborating on the criteria used for study inclusion/exclusion and how disagreements among reviewers were resolved to ensure transparency. I noticed that little was mentioned regarding the exclusion criteria.

Response 2: Thank you very much for your comment. We used the JBI checklist to assess the quality of the studies. We've added lines 139-143. Our choice was a synthesis of all the nursing interventions that the selected studies described as appropriate, i.e. those that promote the empowerment of mastectomised women for self-care. The reader has access to all the interventions reported in the selected studies and can choose, if they wish, to implement the one or ones they consider most appropriate in the context of their clinical practice. We have added lines 154-158 for clarification. We've added lines 110-112 to clarify.

Comment 3: 

  1. Presentation and Completeness
  • Expand the data synthesis section to detail how interventions were categorised and synthesised across studies. Providing a clearer narrative on the main themes and types of interventions would improve comprehensiveness.

Response 3: Thank you very much for your comment. We've expanded the narrative in more detail. We've added lines 159-169; 175-178.

Comment 4:

  1. Language and Style
  • Minor language editing is recommended to improve clarity and style throughout the manuscript. This will enhance readability and ensure your findings are conveyed effectively.
  • DLA: there is a lack of explanation on the term DLA, of which I presumed it stands for Daily living activities. If so, the more commonly used term should be Activities of Daily Living (ADL).
  • It is rather unconventional for the Limitation to come after the Conclusion, since conclusion implies the end of manuscript.

In summary, I appreciate the importance of your study and believe that addressing these points will significantly strengthen your manuscript, making it more robust and valuable to clinicians and researchers alike. I look forward to your revised submission.

Response 4: Thank you very much for your comment and warning. We put ADL on lines 183 and 279.
We put the limitations before the conclusion.

Thank you very much for your work.
Best regards.

Reviewer 4 Report

Comments and Suggestions for Authors

Thank you for the opportunity to read and review this manuscript. This review presents a lot of issues in the methodology and reporting that must be addressed before considering for publication. I think that authors should be more adherent to the JBI Manual and consistent in the reporting. 

In the title, define the type of review. Revise the aim with "to synthesise" or "identify" and not to map. Define the type of design. In the methods, you should describe the inclusion criteria. The number of manuscripts is a result. Why haven't you also searched in Scopus and WoS? The analysis in the methods has not been described. In the conclusion, you should insert the implications.

Introduction

Has the rehabilitation nurse been implemented before? Please report some of the manuscript about this role and the implications of this figure for organisations and patients. What are the differences between the breast nurse and the rehabilitation nurse? I think that you should introduce the role of nurses in breast cancer and sustain the role and precedent manuscripts about the implementation of specialised and dedicated nurses. PROSPERO registration or other dataset is missing.

In the introduction, the aim is missing.

Lines 69-72 are a method.s

The study design is not well-defined. Insert the PRISMA guideline for this review design and charge as a supplementary file. 

Research must also be conducted in Scopus and WoS. You have not considered nursing databases that can influence your results.

The review question is duplicated in the methods and the introduction.

Move the inclusion criteria after the PCC strategy. Please be consistent with inclusion and exclusion criteria. You have poorly described them. 

Reviews are not allowed inside a review; if you do not conduct an overview of reviews.

Search strategy with related results for each dataset must be included as a supplementary file. The search strategy is poor and lacks other terms and uses of the question marks.

Selecting the studies, the inclusion criteria used for the screening are unclear.

The diagram is a result!! In the methodology, you must describe how you have conducted the review. 

Quality appraisal is missing; according to the JBI Manual, you must perform.

Data synthesis is not well described because you have not described how you have conducted it.

Update the review to 2025.

Results

The level of evidence appeared in the table1 but has not been described when and who performed, and on which fundamentals. 

The extraction of the interventions is not clear. How have these been extracted and categorised?

Author Response

Dear Reviewer 4,

Thank you very much for your very useful comments. We will respond to them and we think that the manuscript has been improved by your suggestions. We will respond point by point in order to improve the manuscript. Please let us know if any further changes are needed.

Comment 1: 

Introduction

Has the rehabilitation nurse been implemented before? Please report some of the manuscript about this role and the implications of this figure for organisations and patients. What are the differences between the breast nurse and the rehabilitation nurse? I think that you should introduce the role of nurses in breast cancer and sustain the role and precedent manuscripts about the implementation of specialised and dedicated nurses. PROSPERO registration or other dataset is missing.

Response 1: Thank you very much for your pertinent comment. We've revised the introduction and added lines 62-76 and the PROSPERO register has been added in lines 89-90.

Comment 2: In the introduction, the aim is missing.

Response 2: Thank you very much for your important comment. We've added lines 87-88.

Comment 3: Lines 69-72 are a methods

Response 3: Thank you very much for your comment. We've deleted those lines.

Comment 4: The study design is not well-defined. Insert the PRISMA guideline for this review design and charge as a supplementary file. 

Response 4: Thank you very much for your comment. We have submitted the PRISMA Checklist. 

Comment 5: Research must also be conducted in Scopus and WoS. You have not considered nursing databases that can influence your results.

Response 5: Thank you very much for your comment. It was our methodological choice. In fact, our choice may influence the results and we understand your reservations. 

Comment 6: The review question is duplicated in the methods and the introduction.

Response 6: Thank you very much for your comment. We've removed the question in the introduction.

Comment 7: Move the inclusion criteria after the PCC strategy. Please be consistent with inclusion and exclusion criteria. You have poorly described them. 

Response 7: Thank you very much for your comment. We have moved the inclusion criteria. We have clarified the criteria described in lines 107-112.

Comment 8: Selecting the studies, the inclusion criteria used for the screening are unclear.

Response 8: Thank you very much for your comment. Could you please clarify your comment? We've tried to understand the issue.

Comment 9: The diagram is a result!! In the methodology, you must describe how you have conducted the review. 

Response 9: Thank you very much for your comment. We've put the diagram in the results.

Comment 10: Quality appraisal is missing; according to the JBI Manual, you must perform.

Response 10: Thank you very much for your comment. We have clarified this issue by adding lines 139-143.

Comment 11: Data synthesis is not well described because you have not described how you have conducted it.

Response 11: Thank you very much for your comment. We have clarified the data summary by adding lines 154-169; 175-178.

Comment 12: 

Results

The level of evidence appeared in the table1 but has not been described when and who performed, and on which fundamentals.

Response 12: Thank you very much for your comment. We have clarified this issue by adding lines 172-174.

Comment 13: The extraction of the interventions is not clear. How have these been extracted and categorised?

Response 13: Thank you very much for your comment. We've added lines 175-178.

Comment 14:  In the conclusion, you should insert the implications.

Response 14: Thank you very much for your pertinent comment. We've added lines 314-338.

Thank you very much for your work.
Best regards.

Round 2

Reviewer 1 Report

Comments and Suggestions for Authors

The manuscript has been significantly improved since version one, most likely also thanks to the comments made by the other reviewers.

Author Response

Dear Reviewer 1,

Thank you very much for your comment.

Best regards.

Reviewer 3 Report

Comments and Suggestions for Authors

Dear authors,

Thank you very much for the revision and the opportunity to review your manuscript again. I have noted your revision, and the manuscript is now reading more robustly. However, there are still several minor revisions which I would like to suggest: 

  1. Lines 55 to 90: please refrain from using single-sentence paragraphs, and there are three such paragraphs within six lines.
  2. Lines 107 to 109 and Lines 288 to 290: Please elaborate why only the past 5 years of literature were searched; the explanation of "recent" literature is insufficient. More illustrative reasons, such as a drastic change or reform in the healthcare system or clinical practice, could potentially be a reason if the intention is not to collude older or outdated practices with the current ones. However, I find it perplexing that the decision to include literature from only this 5-year time frame was made. Yet, it is addressed as a study limitation.  This is illogical and will remain the biggest flaw of your manuscript. I cannot help but ask, "Why would you choose to do this?!" 
  3. The whole section of lines 314 to 338 should be under discussion instead of conclusion. 
  4. Figure 2 appeared blurry and illegible. 

Author Response

Dear Reviewer 3,

Thank you very much for your considerations and comments, which have been very useful as we continue to improve the manuscript. We will respond to them and we think that the manuscript has been improved by your suggestions. Please let us know if any further changes are needed.

Comment 1: 

Lines 55 to 90: please refrain from using single-sentence paragraphs, and there are three such paragraphs within six lines.

Response 1:

Thank you very much for your comment. Indeed, you're right. We went over this issue in these lines and in other parts of the manuscript.

Comment 2: 

Lines 107 to 109 and Lines 288 to 290: Please elaborate why only the past 5 years of literature were searched; the explanation of "recent" literature is insufficient. More illustrative reasons, such as a drastic change or reform in the healthcare system or clinical practice, could potentially be a reason if the intention is not to collude older or outdated practices with the current ones. However, I find it perplexing that the decision to include literature from only this 5-year time frame was made. Yet, it is addressed as a study limitation.  This is illogical and will remain the biggest flaw of your manuscript. I cannot help but ask, "Why would you choose to do this?!" 

Response 2: 
Thank you very much for your comment. There have indeed been recent changes in Portugal, with a greater increase in the role of the rehabilitation nurse for various reasons. We'll clarify this in lines 106-111. However, we understand your criticisms.

Comment 3:

The whole section of lines 314 to 338 should be under discussion instead of conclusion.

Response 3: 

Thank you very much for your comment. We've moved those lines. They are now lines 294-318.

Comment 4: 

Figure 2 appeared blurry and illegible. 

Response 4: 

Thank you very much for the alert. We have improved the sharpness of the text in the image. If the reviewer considers that the image needs to be improved, we will look for another image editor.

Thank you very much for your work.

Best regards.

Reviewer 4 Report

Comments and Suggestions for Authors

Thank you for your manuscript improvements.

Lines 154-169 are methodology because you describe how you have synthesised the results. 

In the methods section, data synthesis, you have not declared how you have identified this intervention area, e.g. recurrent or most frequent words in the section of the manuscript of the results and discussion. 

Lines 172-174 are duplicated; remove them. Describe the quality of the study.

In Table 1, the quality of the study is not of JBI appraisal but another methodology. For each research, you have to declare each point assigned by the researchers on the instrument quality evaluation. You have declared the quality appraisal, but not included in the results. 

Table 1 and 2 are quite similar. The information of each single intabulated study must be charged as a supplementary files, with a table including authors, years, title, aim, study design, methods, results, conclusion. You must adhered to the JBI manual, as you declared have followed for this review.

Please also see that the PCC methodology is for SCOPING review and not systematic. Your review is a scoping review, due to the strategy and the results synthesis choosen.

The search strategy has not been revised. You have to insert the exact search strategy with the use of * and ". Besides, in a supplementary file, you have to report each search with the search strategy, results obtained for each dataset on the day of the conduct of this review. We can not replicate your search if you simply state until 2024. You have to declare the day and month. A revision must be replicable and, in future, can be updated starting from the day after your search. 

Author Response

Dear Reviewer 4,

Thank you very much for your considerations, comments and all the work you have put into reviewing the manuscript, as they have been very useful in terms of continuing to improve the manuscript. We will respond to them and we think that the manuscript has been improved by your suggestions. Please let us know if any further changes are needed.

Comment 1: Lines 154-169 are methodology because you describe how you have synthesised the results. 

Response 1: Thank you very much for your pertinent comment. We've put this text in the methodology. It's lines 152-167.

Comment 2: In the methods section, data synthesis, you have not declared how you have identified this intervention area, e.g. recurrent or most frequent words in the section of the manuscript of the results and discussion. 

Response 2: Thank you very much for your comment. We have clarified how we summarised the data by adding lines 140-146.

Comment 3: Lines 172-174 are duplicated; remove them. Describe the quality of the study.

Response 3: Thank you very much for the important warning. We've removed the lines.

Comment 4: In Table 1, the quality of the study is not of JBI appraisal but another methodology. For each research, you have to declare each point assigned by the researchers on the instrument quality evaluation. You have declared the quality appraisal, but not included in the results. 

Response 4: Thank you very much for your important comment. We have clarified the quality of the studies by adding lines 180-182. Does the reviewer want us to send the JBI checklist as supplementary material?

Comment 5: Table 1 and 2 are quite similar. The information of each single intabulated study must be charged as a supplementary files, with a table including authors, years, title, aim, study design, methods, results, conclusion. You must adhered to the JBI manual, as you declared have followed for this review.

Response 5: Thank you very much for your pertinent comment. We have submitted the information from each study as supplementary material. We would like to ask if we can keep tables 1 and 2 in the manuscript? We ask this because we would like to avoid potential conflict with other reviewers. If any other changes are necessary, please let us know.

Comment 6: Please also see that the PCC methodology is for SCOPING review and not systematic. Your review is a scoping review, due to the strategy and the results synthesis choosen.

Response 6: Thank you very much for your pertinent comment. We've changed the title.

Comment 7: The search strategy has not been revised. You have to insert the exact search strategy with the use of * and ". Besides, in a supplementary file, you have to report each search with the search strategy, results obtained for each dataset on the day of the conduct of this review. We can not replicate your search if you simply state until 2024. You have to declare the day and month. A revision must be replicable and, in future, can be updated starting from the day after your search. 

Response 7: Thank you very much for your comment. We have reformulated the search strategy in lines 119-121. We have declared the publication dates of those used in the search on line 104. We have sent a supplementary file reporting each search.

Thank you very much for your work.
Best regards.